# Dietary Protein Requirement Threshold and Micronutrients Profile in Healthy Older Women Based on Relative Skeletal Muscle Mass

**DOI:** 10.3390/nu13093076

**Published:** 2021-09-01

**Authors:** Praval Khanal, Lingxiao He, Hans Degens, Georgina K. Stebbings, Gladys L. Onambele-Pearson, Alun G. Williams, Martine Thomis, Christopher I. Morse

**Affiliations:** 1Musculoskeletal Science and Sports Medicine Research Centre, Department of Sport and Exercise Sciences, Manchester Metropolitan University, Manchester M15 6BH, UK; lingxiao.he@hotmail.com (L.H.); G.Stebbings@mmu.ac.uk (G.K.S.); G.Pearson@mmu.ac.uk (G.L.O.-P.); a.g.williams@mmu.ac.uk (A.G.W.); c.morse@mmu.ac.uk (C.I.M.); 2Department of Movement Sciences, Physical Activity, Sports & Health Research Group, KU Leuven, 3001 Leuven, Belgium; martine.thomis@kuleuven.be; 3Department of Life Sciences, Manchester Metropolitan University, Manchester M15 6BH, UK; h.degens@mmu.ac.uk; 4Institute of Sport Science and Innovations, Lithuanian Sports University, LT-44221 Kaunas, Lithuania; 5Pharmacy of Targu Mures, University of Medicine, 540142 Targu Mures, Romania; 6Institute of Sport, Exercise and Health, University College London, London W1T 7HA, UK; 7Applied Sports Science Technology and Medicine Research Centre, Faculty of Science and Engineering, Swansea University, Swansea SA1 8EN, UK

**Keywords:** pre-sarcopenia, musculoskeletal health, protein

## Abstract

Although multiple nutrients have shown protective effects with regard to preserving muscle function, the recommended amount of dietary protein and other nutrients profile on older adults for maintenance of high muscle mass is still debatable. The aims of this paper were to: (1) identify dietary differences between older women with low and high relative skeletal muscle mass, and (2) identify the minimal dietary protein intake associated with high relative skeletal muscle mass and test the threshold ability to determine an association with skeletal muscle phenotypes. Older women (*n* = 281; 70 ± 7 years, 65 ± 14 kg), with both low and high relative skeletal muscle mass groups, completed a food questionnaire. Skeletal muscle mass, fat-free mass (FFM), biceps brachii thickness, *vastus lateralis* anatomical cross-sectional area (VL_ACSA_), handgrip strength (HGS), maximum elbow flexion torque (MVC_EF_), maximum knee extension torque (MVC_KE_), muscle quality (HGS/Body mass), and fat mass were measured. Older women with low relative skeletal muscle mass had a lower daily intake of protein, iodine, polyunsaturated fatty acid (PUFA), Vit E, manganese, milk, fish, nuts and seeds (*p* < 0.05) compared to women with high relative skeletal muscle mass. The minimum required dietary protein intake for high relative skeletal muscle mass was 1.17 g/kg body mass/day (g/kg/d) (sensitivity: 0.68; specificity: 0.62). Women consuming ≥1.17 g/kg/d had a lower BMI (B = −3.9, *p* < 0.001) and fat mass (B = −7.8, *p* < 0.001), and a higher muscle quality (B = 0.06, *p* < 0.001). The data indicate that to maintain muscle mass and function, older women should consume ≥1.17 g/kg/d dietary protein, through a varied diet including milk, fish and nuts that also contain polyunsaturated fatty acid (PUFA) and micronutrients such as iodine, Vit E and manganese.

## 1. Introduction

One of the factors contributing to age-related skeletal muscle degeneration is poor nutrient intake [1]. In extreme conditions, such as anorexia of ageing, inadequate food intake has been associated with impaired physical performance and an increased risk of disability among older community-dwellers [2]. The effects of caloric deficit in these conditions [3] may be further aggravated by a lack of dietary nutrients linked to anabolic or anti-catabolic process in the muscle. Identification of the nutrients profile in the diet of independently living older adults with limited, if any, physical impairments may help inform dietary interventions to prevent premature disability in the older adults.

Current studies of dietary intake and skeletal muscle function in the older adults mainly focus on protein intake [4,5], which may be due to positive associations between muscle protein synthesis and dietary protein intake [6]. Despite the beneficial effect of protein intake on protein synthesis, the minimum recommended daily allowance (RDA) of 0.8 g/kg body mass/day (g/kg/d) for an older person is still a matter of debate [7]. For instance, some studies report that a protein intake between 1.0–1.2 g/kg/d is appropriate for musculoskeletal health [8,9,10], while others recommend >1.2 g/kg/d to combat sarcopenia [11,12] and the British Nutrition Foundation (BNF) even suggested that the RDA for the older adults is as low as 0.75 g/kg/d [13].

Low muscle mass is an indication of the initial stage for the loss of independence among the older population and has been linked to adverse outcomes such as increased risk of physical disability [14] and loss of mobility [15]. Identification of a dietary protein intake threshold that is minimally required to maintain high skeletal muscle mass may be useful for older people to prolong their independence. In addition to protein, micronutrients such as Vit D, calcium, manganese, iron, and zinc are particularly important—either in the process of energy metabolism acting as co-factors, or helping in providing antioxidants [16,17,18,19,20,21]. Indeed, associations of these dietary micronutrients with muscle mass, performance, and sarcopenia have been reported [22,23,24,25], but other studies did not see such associations [26,27].

Investigating the beneficial effect of protein and other nutrients on muscle phenotypes has utilized several approaches previously, with some studies identifying direct associations between nutrients and muscle phenotypes while some identified nutrients profiles of healthy elderly individuals [28,29,30,31]. Although there are multiple approaches, there is no study that has established the dietary protein intake required for maintaining the high level of skeletal muscle mass based on pre-sarcopenic (low relative skeletal muscle mass) threshold. An identification of the dietary protein requirement for the older population based on the threshold of relative skeletal muscle mass is important as it could influence individual physical independence, and progression to sarcopenia. For example, a lower muscle mass has been linked with a decline in parameters related to physical independence such as functional impairment (walking and chair stands) [14], balance [32] and respiratory strength [33] and other conditions such as development of metabolic syndrome [34], incidence of hypertension [35] and likelihood of physical disability [14]. 

To combat movement impairment during ageing, cognitive behaviour and exercise interventions [36] are recommended with, in particular, resistance training [37] being effective in restoring muscle mass and reversing sarcopenia. It should be noted that a change in diet, which requires the establishment of recommended dietary protein intake, and identifying micronutrients beneficial for musculoskeletal health represents a lifestyle change that requires no specialist equipment or time commitment. With this in mind, the aims of the present study were: (1) to compare the intake of dietary nutrients and food items between older women with low and high relative skeletal muscle mass; and (2) to identify the dietary protein threshold required to delineate older women with low vs. high level relative skeletal muscle mass and test its ability to associate with muscle and body composition phenotypes.

## 2. Materials and Methods

### 2.1. Participant Characteristics

Participants in this study were recruited from the “Genetics of Sarcopenia” project (*n* = 307, Manchester Metropolitan University, UK) from which 281 participants (60–91 years) with complete food intake data were included. The recruiting criteria were: (1) Caucasian women, (2) >60 years, (3) no mobility-related problems and (4) no history of muscle or nervous system conditions that might affect independent living. All procedures performed in studies involving human participants were in accordance with governmental agency of the UK and the ethical standards of the institution research committee (Manchester Metropolitan University Ethics Committee; Approval number: 09.02.16 (i)) and with the 1964 Helsinki declaration and its later amendments or comparable ethical standards. All participants provided written informed consent.

The following phenotypes were evaluated: fat mass, fat-free mass (FFM), skeletal muscle mass (SMM), biceps brachii thickness, *vastus lateralis* anatomical cross-sectional area (VL_ACSA_), handgrip strength (HGS), maximum elbow flexion torque (MVC_EF_), maximum knee extension torque (MVC_KE_) and HGS/Body mass. 

### 2.2. Skeletal Muscle Mass Measurement and Low Muscle Mass Definition

With participants lying in a supine position, skeletal muscle mass was estimated using bioimpedance (BIA) analysis (Bodystat 1500MDD, Isle of Man, UK) [38]. Skeletal muscle mass was estimated using a validated equation [39]. The measurement of muscle mass with BIA has been highly correlated with the measures of DEXA (ICC = 0.90) [40].

As body mass will influence the skeletal muscle mass, relative skeletal muscle mass (SMM_r_) was calculated as 100 × SMM/body mass and individuals with SMM_r_ < 22.1% were considered pre-sarcopenic [14]. There remains an ongoing debate on the preferred method of reporting skeletal muscle mass normalised for body mass for identifying lower levels in the older population [41,42]. The 22.1% SMM_r_ in the present study has previously been used for identifying risk of disability [14], the gene variants associated with lower muscle mass (sarcopenia)[38] and balance impairments in the elderly [32]. For the purposes of group comparisons, and to be consistent with the established terminology [42], those participants with a SMM_r_ < 22.1% will be termed “low relative skeletal muscle mass”, those with a SMM_r_ ≥ 22.1% will be termed “high relative skeletal muscle mass” [14]. 

### 2.3. Vastus Lateralis and Biceps Brachii Size

B-mode ultrasonography (7.5 MHz, linear array probe, 38 mm; MyLab Twice, Esaote Biomedical, Genoa, Italy) was used to perform a transverse plane scan at 50% of *vastus lateralis* (VL) muscle length. Ultrasound images were recorded (Adobe Premier, Adobe) and the entire muscle was reconstructed in a single canvas. Digitizing software (ImageJ 1.45, National Institute of Health, Bethesda, MD, USA) was used to measure the *vastus lateralis* anatomical cross-sectional area (VL_ACSA_).

The same ultrasound unit was used to measure the size of the biceps brachii as described previously [43]. With the dominant hand hanging freely by the participant’s side, a sagittal plane scan was performed at 60% length from the proximal end of the humerus. The biceps brachii thickness was recorded as the mean thickness measured at proximal, middle and distal points of the captured image.

The scan was performed by same investigator with high consistency for VL_ACSA_ (ICC = 0.99) and biceps brachii thickness (ICC = 0.98), calculated based on duplicate measures of six participants.

### 2.4. Muscle Strength and Muscle Quality

HGS was evaluated with a handgrip dynamometer (JAMAR plus, JLW Instruments, Chicago, IL, USA). MVC_EF_ and MVC_KE_ were measured with a custom-built dynamometer. The detailed process for the muscle strength measurements are described in our previous work [38,44].

In short, participants squeezed the handgrip dynamometer with maximal strength, with the arm down the side of the body. HGS of the participants was defined as the highest value of the six attempts performed by both hands (three by each hand, alternately, with 10 s interval between trials).

For MVC_EF,_ participants were seated in a custom-built dynamometer and instructed to produce maximum force with the elbow flexed at 120° (0° is a straight position) with their dominant arm. The maximum of three trials was recorded and MVC_EF_ was calculated as:MVC_EF_ = Force × radius length × Cos30°

For MVC_KE_, participants were seated in a custom-built dynamometer and asked to produce maximum force with the knee angle at 120° (straight was considered as 180°) with the dominant leg. The highest of three attempts was recorded and converted to torque as:MVC_KE_ = Force × distance from rotation point of dynamometer to ankle strap × Cos30°

There was high test-retest reliability for both the MVC_EF_ (ICC = 0.95) and MVC_KE_ (ICC = 0.96) in six participants.

Muscle quality was subsequently calculated as HGS/Body mass.

### 2.5. Physical Activity Scale for the Elderly Questionnaire

Self-reported physical activity was obtained using the physical activity scale for the elderly (PASE) questionnaire and evaluated in a one-on-one interview on the same day that muscle parameters were assessed [45]. 

### 2.6. Dietary Assessment

Dietary items and nutrient intake were assessed in a one-on-one interview using the European Prospective Investigation into Cancer and Nutrition Norfolk Food Frequency Questionnaire (EPIC-Norfolk FFQ) [46]. The FFQ asks the food intake frequency of a participant ranging from “never or less than once/month” to “six times per day” for 131 food items. This FFQ has been validated with a 24-h Diet Recall [46] and 7-Day Diet Diary [47] in UK cohorts. Responses were processed using the CAFE program [48], which converted frequency of each food category into a daily nutrient intake.

### 2.7. Statistics

Statistical analyses were performed in SPSS 27.0 and *p* < 0.05 was considered significant in all the analyses. Low relative skeletal muscle mass was defined with the previously established threshold of 22.1% SMM_r_ [14]. The dietary nutrients intake values presented in the current study are given as daily intake (/day), and for protein it is daily intake per kg of body mass (day/kg) unless reported otherwise. The participants were grouped into five age-group categories with intervals of five years. The general characteristics, muscle and diet related variables were first assessed for their normal distribution using a Kolmogorov-Smirnov (*n* > 50) or Shapiro-Wilk (*n* < 50) test when appropriate. When data were normally distributed, analysis of variance (ANOVA) was run; otherwise, a non-parametric Kruskal-Wallis test was performed to compare age groups for general characteristics, muscle phenotypes, and dietary foods and nutrients intake. Post-hoc pairwise comparisons were conducted using Bonferroni corrections when a statistically significant main effect was found. Independent sample t-tests or Mann-Whitney tests (whichever was appropriate) were used to compare low and high SMM_r_ groups. For parametric data mean ± SD are reported, while median (IQR) are reported for non-parametric data.

Receiver operating characteristics (ROC) curve analysis was performed to identify the dietary protein intake threshold that distinguished older women with low SMM_r_ from older women with high SMM_r_. The minimally required protein threshold was defined as the value that had both higher sensitivity and specificity values, and subsequently participants were classified into two groups based on the established threshold. Pearson’s Chi-square test was conducted to compare the frequency of participants with low SMM_r_ between the groups meeting and not meeting the identified threshold of dietary protein intake. Associations of dietary protein based on this established threshold (coded as a binary categorical variable) with BMI, fat mass and skeletal muscle phenotypes of size, strength and quality were investigated using linear regression adjusted for age, energy-intake, and physical activity.

## 3. Results

Older participants and low SMM_r_ participants were characterized by lower SMM, MVC_KE_ and muscle quality (Table 1).

### 3.1. Dietary Food Items and Nutrient Intake Pattern of Participants with Age-Group Categories and Pre-Sarcopenia

There was no difference in daily energy or nutrient intake between age group categories (Table 2).

Compared to participants with a high SMM_r_, those with low SMM_r_ had lower intake of protein relative to body mass (−18.0%, *p* < 0.001), iodine (−12.4%, *p* = 0.015), manganese (−13.4%, *p* = 0.009), Vit E (−10.0%, *p* = 0.047), and Poly-unsaturated fatty acid (PUFA) (total (−13.1%, *p* = 0.023)). Other dietary nutrient intakes were not different between the high and low SMM_r_ groups (Table 2).

Dietary food items intake did not differ between age groups (Table 3). Compared to participants with a high SMM_r_, those with low SMM_r_ had lower daily intake of milk and milk products, fish and fish products, nuts and seeds, with higher daily intake of meat and meat products (*p* < 0.05, Table 3).

### 3.2. Minimally Required Dietary Protein Intake for a High Relative Muscle Mass

Based on a 22.1% low SMM_r_ threshold, AUC of the model that predicted the daily dietary protein intake was 0.70. The minimal required dietary protein intake for high SMM_r_ was 1.17 g/kg/d (sensitivity: 0.68; specificity: 0.62, Figure 1). Of all participants, 35.9% (*n* = 101) did not achieve the threshold for dietary protein intake of 1.17 g/kg/d.

A greater proportion of individuals identified as having low SMM_r_ (59.5%, *n* = 22) failed to meet the dietary protein intake threshold of 1.17 g/kg/d compared with 32.4% in the high SMM_r_ (χ^2^ = 10.2, *p* = 0.001).

The consumption of dietary protein above the established low SMM_r_ threshold of 1.17 g/kg/d was associated with lower BMI (B = −3.88, *p* < 0.001), fat mass (B = −7.84, *p* < 0.001) and biceps brachii thickness (B = −0.15, *p* < 0.001) and higher HGS/Body mass (B = 0.063, *p* < 0.001, Table 4).

## 4. Discussion

The present cross-sectional study adopted two approaches to assess the importance of dietary intake on muscle phenotypes in older women. In brief these were: (1) the comparison of dietary nutrients and food intake between older adults with low and high SMM_r_ groups; and (2) a comparison of those participants consuming above and below our calculated protein threshold of 1.17 g/kg/d. Our main findings were that there was no difference in the dietary intake between 60- to 91-year-old women, but intake of protein, iodine, Vit E, manganese and PUFA was higher in women with high SMM_r_ compared to those with low SMM_r_. The current study also identified a threshold of 1.17 g/kg/d protein required for the maintenance of high SMM_r_, where consuming more protein than this threshold was associated with lower BMI, lower fat mass, and higher muscle quality in older women. Our findings suggest that nutrients associated with processes such as muscle protein synthesis, energy metabolism, or affecting gene expression in skeletal muscle are particularly important for high skeletal muscle mass, muscle strength, and muscle quality in older women.

### 4.1. Dietary Nutrients Intake Difference with Age-Groups and Pre-Sarcopenia Status

There are numerous studies that report that appetite and energy intake decrease with ageing [8,49]. However, the current study observed that there was no impact of age on absolute calorific or nutrient intake in our sample of older women. The possible cause of such an absence in energy intake difference between age-groups in our study could be the inclusion of a sample within a relatively narrow age range (60–91 years) compared to previous studies that reported energy intake differences in younger and older populations [49,50,51].

The lower intake of proteins, and PUFA, and some micronutrients, such as iodine, manganese and Vit E in the low SMM_r_ group than high SMM_r_ group may be attributable to both a lower food intake and a diet lacking sufficient quality to meet the RDA of some nutrients (e.g., lower intake of nuts, milk, and fish specifically). The findings from the current study show that older women with low SMM_r_ are characterized by low intake of certain dietary nutrients and that this is consistent with evidence that lower nutrient intake and diet quality and a less-varied diet are associated with ageing and poor muscle health [52,53]. Our study is in line with previous studies that suggest the necessity of consuming a variety of foods to obtain all of the nutrients that are required to maintain greater muscle mass/strength among an older population [19,54]. In the present study, we observed the consumptions of higher amounts of protein, iodine, polyunsaturated fatty acids (PUFA), Vit E and manganese are favourable for maintenance of muscle level above the pre-sarcopenic threshold using SMM_r_. It is possible that food types may underlie the lower nutrient intake, where the high SMM_r_ group had a diet richer in fish, milk, nuts and seeds, considered a high diet quality [55,56,57,58,59,60,61,62]. The association of higher dietary intake of protein with better muscle phenotypes in the present study could be explained by the reported positive link between dietary protein intake and muscle protein synthesis [6]. Although there is some evidence that the associated micronutrients may have some muscle-preserving actions (e.g., Vit E through antioxidant effects [63]), or affect thyroid hormone dependent gene expression in skeletal muscle (e.g., Iodine) [64], or affect energy metabolism processes (e.g., Manganese) [65], it is not possible to identify which of the micronutrients is the discriminating factor between the diets and muscle mass differences of our low and high SMM_r_ groups.

### 4.2. Dietary Protein Threshold for Low Relative Skeletal Muscle Mass, Skeletal Muscle Phenotypes and Body Composition

ROC analysis revealed that 1.17 g/kg/d dietary protein intake is the minimally required amount to maintain high SMM_r_. This is higher than the current consensus of 0.8 g/kg/d for the maintenance of skeletal muscle health among the older [10] and the protein intake recommended by the BNF (0.75 g/kg body mass/day) [13]. Yet, our derived threshold of protein intake for the maintenance of high SMM_r_ is close to the suggested range of 1.0–1.2 g/kg/d as appropriate for muscle health in an older population reported by others [8,9,10], and aligns with other studies reporting that 0.8 g/kg body mass/day is not sufficient to maintain muscle function in older adults [10,53,66]. Given that during acute and chronic condition an even higher protein intake (>1.2 g/kg/d) is recommended in older adults [10], we suggest that an RDA of 0.75 or 0.8 g/kg/d is insufficient to sustain muscle health in older women.

Here we also observed that participants with a daily protein intake <1.17 g/kg/d had a higher BMI and fat mass and lower muscle quality than those with an intake above this threshold. Given that there were only few differences in dietary nutrient intake between women with a low SMM_r_ and those with a high SMM_r_, it seems that a diet with higher than our identified protein threshold content is indicative of one that can facilitate lower levels of body fat and BMI, and higher muscle quality. This is in line with some previous reports linking obesity-related indices such as a high BMI and fat mass to lower protein intake [67,68], but not all [69]. The larger biceps brachii thickness with consumption of protein <1.17 g/kg/d threshold is counterintuitive, but may be attributed to a higher body fat, and likely to intramuscular fat infiltration in the low protein consuming group, a known source of overestimating muscle mass when using ultrasonography [70]. Based on the differences in dietary protein intake between groups having low SMM_r_ and high SMM_r_, and the association of dietary protein intake ≥1.17 g/kg/d with favourable body composition and skeletal muscle phenotypes, we suggest that 1.17 g/kg/d dietary protein intake is minimally required to maintain high muscle mass associated with ageing in women.

It cannot be overlooked that the contribution of a heavier body mass, and higher adiposity could contribute to the observed results that lower SMM_r_ group have lower relative protein intake. Of course, we acknowledge that there is a cachexic role of adipokines in the older population that would likely contribute to the lower muscle mass in this group [71]. However it is likely through lower quality diet and lower protein intake that the higher adiposity was observed (with other external factors such as physical activity being equal). As with observational studies such as the present, it is not possible to identify a causal relation to the higher relative protein intake and high muscle mass, but it certainly contributes to the growing evidence that diet quality and relative protein content (particularly above a threshold) are important for the maintenance of muscle mass in the older adults.

### 4.3. Limitations

The FFQ has a number of shortcomings which are common with all recall questionnaires, and self-reporting approaches [72], such as underreporting compared to direct observations or double-labelled water (for caloric intake). It is unlikely however that our observations are invalidated using FFQ, but some of the more nuanced detail regarding micronutrient intake may have been missed. One example is in Vit D, where the FFQ captures Vit D2 intake only, but does not provide information on Vit D3 intake. Although dietary protein, and some micronutrients are important in distinguishing between low and high SMM_r_ groups in the present study, this is not an exhaustive list, particularly given the known Vit D deficiency which may persist within older women in the UK [73]. Furthermore, the associations of skeletal muscle phenotypes and nutrients intake could be better understood if the nutrients are measured directly from a blood sample unlike the questionnaire used in the present study. The present study also did not assess nutrient supplements the participants might have consumed in the past year.

In addition to above, the recent literature emphasizes the use of SMI (skeletal muscle mass adjusted with height squared) instead of SMM_r_ for evaluating individual’s muscle mass level. Although variable differences in prevalence of low muscle mass could be anticipated with SMI and SMM_r_ thresholds [38] and thus the possible outcome measures in investigation, the authors present the findings in relation to relevance of relative skeletal muscle mass for physical independence and clinical conditions in older population [14,32,33,34,35]. PASE is a valid and reliable assessment tool for the classification of physical activity in the older population [74], with approximation to quantitative methods of assessment [75]. Despite this, we are aware that some element of overestimating higher intensity physical activity domains does occur in the older population [76]. This could account for a lack of physical activity difference between high and low SMM_r_ groups. However, this doesn’t invalidate our primary conclusion that diet, and protein intake are lower in those with low SMM_r_. We cannot confirm whether the lack of difference in physical activity would persist if more sensitive physical activity measurement tools were adopted.

## 5. Conclusions

The present study showed that a 1.17 g/kg/day dietary protein intake is minimally required to maintain high muscle mass level in older women. We also identified that a higher consumption of PUFA and micronutrients (iodine, polyunsaturated fatty acid, Vit E, and manganese) along with food items such as nuts and seeds, milk and milk products, and fish and fish products are associated with greater skeletal muscle mass and strength. We therefore suggest the importance of increasing the variety and amount of dietary nutrients in older adults diets, particularly protein and a number of micro-nutrients associated with diets of high quality (e.g., incorporating fish and nuts), which may help maintain high muscle mass and function in older people.

## Figures and Tables

**Figure 1 nutrients-13-03076-f001:**
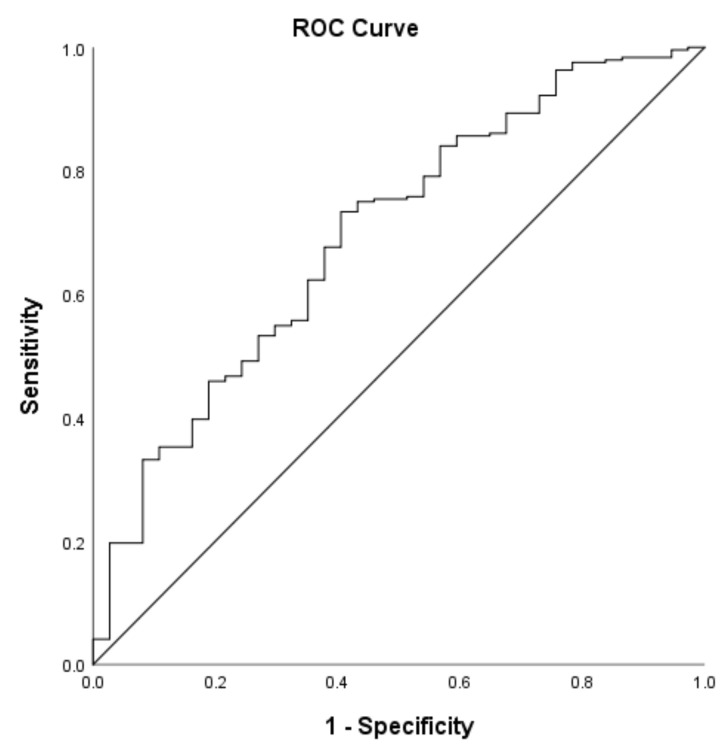
ROC curve showing the association of protein-intake with 22.1% SMM_r_ low muscle mass threshold.

**Table 1 nutrients-13-03076-t001:** General characteristics and skeletal muscle phenotypes of participants according to age group and muscle mass status.

General Characteristics	All Participants *n* = 281	60–64 Years *n* = 39	65–69 Years *n* = 98	70–74 Years *n* = 94	75–79 Years *n* = 34	80–91 Years *n* = 16	*p*-Value (Age Groups)	Low Relative Skeletal Muscle Mass *n* = 37	High Relative Skeletal Muscle Mass *n* = 244	*p*-Value (Relative Skeletal Muscle Mass Groups)
Age (years)	70 (7)	62 (2)^2,3,4,5^	68 (2) ^1,3,4,5^	73 (3) ^1,2,4,5^	77 (3) ^1,2,3^	82 (8) ^1,2,3^	<0.001	70 (6)	70 (7)	0.682
Body mass (kg)	65 (14)	64.8 (16.0)	66.3 (12.1)	65.6 (12.2)	61.8 (17.3)	69.2 (13.1)	0.108	76.1 (20.9)	64.8 (11.8)	<0.001
Height (m)	1.60 (0.08)	1.60 (0.06)	1.60 (0.07)	1.60 (0.08)	1.58 (0.09)	1.56 (0.13)	0.032	1.60 (0.08)	1.60 (0.08)	0.788
BMI (kg/m^2^)	25.4 (4.5)	24.7 (4.8) ^4^	25.5 (4.4) ^4^	25.4 (5.1)	24.8 (3.9) ^1,2^	27.1 (5.9)	0.029	30.3 (7.8)	24.9 (4.1)	<0.001
SMM_r_	25.8 ± 4.0	27.6 (5.8) ^3,5^	25.9 (4.6)	25.2 (5.1) ^1^	26.0 (3.9)	23.9 (5.2) ^1^	0.007	20.1 (2.2)	26.2 (4.3)	<0.001
SMM (kg)	16.9(3.1)	17.1(2.5) ^3,4,5^	17.6 (3.3) ^3,4^	16.6 (3.0) ^1,2^	16.5 (2.8) ^1,2^,	15.8 (3.3) ^1^	0.013	15.0 (4.2)	17.1(2.8)	<0.001
Fat mass (kg)	27.1 (8.6)	25.1 (11.4) ^5^	27.3 (8.3)	27.4 (8.3)	25.8 (8.0)	31.4 (8.5) ^1^	0.012	37.6 (12.3)	26.5 (7.7)	<0.001
RMR (kcal)	1268 ± 110	1288 ± 111 ^4^	1291 ± 112 ^4^	1262 ± 98	1213 ± 121 ^1,2^	1235 ± 89	0.003	1288 ± 110	1265 ± 110	0.246
Biceps brachii thickness (cm)	1.71 (0.41)	1.80 (0.48)	1.73 (0.43)	1.69 (0.43)	1.87 (0.50)	1.64 (0.33)	0.482	1.74 (0.43)	1.69 (0.41)	0.521
VL_ACSA_ (cm^2^)	16.4 ± 3.3	18.6 ± 3.1 ^2,3,4,5^	16.6 ± 3.4 ^1^	16.2 ± 3.2 ^1^	15.3 ± 2.9 ^1^	14.7 ± 2.5 ^1^	<0.001	17.0 ± 3.7	16.4 ± 3.3	0.306
HGS (kg)	30.0 ± 4.9	32.5 (6.1) ^4,5^	30.8 (6.3) ^4,5^	29.9 (5.1) ^5^	27.4 (3.9) ^1,2^	23.7 (7.8) ^1,2,3^	<0.001	28.7 ± 4.6	30.2 ± 5.0	0.081
MVC_EF_ (N·m)	28.3 (9.1)	31.1 (11.6) ^5^	28.9 (8.7) ^5^	28.4 (8.5)	27.6 (8.4)	24.7 (7.8) ^1,2^	0.020	26.7 (9.0)	28.5 (9.1)	0.140
MVC_KE_ (N·m)	64.7 ± 21.3	72.3 (27.8) ^4,5^	66.4 (31.5) ^5^	65.6 (27.6)	54.4 (33.2) ^1^	47.0 (32.1) ^1,2^	<0.001	53.1 (26.2)	65.9 (28.3)	0.022
HGS/Body mass (kg/kg)	0.46 ± 0.09	0.51 ± 0.10 ^3,5^	0.47 ± 0.10 ^5^	0.46 ± 0.09 ^1,5^	0.46 ± 0.08 ^5^	0.36 ± 0.08 ^1,2,3,4^	<0.001	0.38 ± 0.07	0.47 ± 0.09	<0.001
PASE	158 ± 50	156 (67)	153 (69)	161 (56)	158 (52)	122 (59)	0.209	145 ± 51	160 ± 50	0.089

Abbreviations: BMI, Body mass index; SMM_r_, Relative skeletal muscle mass; SMM, Skeletal muscle mass; RMR, Resting metabolic rate; VL_ACSA_, Vastus lateralis anatomical cross-sectional area; HGS, Handgrip strength; MVC_EF_, Maximum elbow flexion torque; MVC_KE_, Maximum knee extension torque; PASE, Physical activity scale for the elderly. ^1,2,3,4,5^ indicate difference from 60–64 years, 65–69 years, 70–74 years, 75–79 years and 80+ years, respectively at *p* ≤ 0.05.

**Table 2 nutrients-13-03076-t002:** Dietary nutrient intake of participants according to age group and muscle mass status.

Dietary Nutrients Intake (Per Day)	All Participants *n* = 281	60–64 Years *n* = 39	65–69 Years *n* = 98	70–74 Years *n* = 94	75–79 Years *n* = 34	80–91 Years *n* = 16	*p*-Value (Age Groups)	Low Relative Skeletal Muscle Mass *n* = 37	High Relative Skeletal Muscle Mass *n* = 244	*p*-Value (Relative Skeletal Muscle Mass Groups)
Energy(kcal)	1735 (552)	1821 (498)	1696 (527)	1728 (623)	1850 (565)	1746 (684)	0.476	1685 (545)	1739 (568)	0.217
Carbohydrate (g)	199 (71)	206 (86)	192 (65)	200 (80)	218 (57)	202 (70)	0.336	183 (68)	201 (68)	0.206
Carbohydrate (%TEI)	43.2 (8.8)	43.5 (8.8)	42.5 (8.6)	43.6 (8.1)	43.1 (10.0)	42.8 (10.0)	0.816	43.3 (8.5)	43.2 (8.8)	0.707
Fat-total (g)	66 (28)	66 (23)	64 (27)	62 (26)	70 (26)	68 (31)	0.207	64 (24)	66 (30)	0.272
Fat-total (%TEI)	34.2 (7.8)	34.3 (10.3)	34.3 (7.2)	33.7 (7.5)	35.3 (7.7)	36.6 (8.1)	0.482	33.7 (8.6)	34.2 (7.8)	0.533
Protein (g)	86 (31)	81 (39)	85 (30)	86 (28)	89 (32)	86 (33)	0.848	86 (32)	86 (31)	0.778
Protein (g/kg/d)	1.31 (0.55)	1.26 (0.51)	1.29 (0.57)	1.31 (0.55)	1.44 (0.63)	1.38 (0.58)	0.404	1.08 (0.49)	1.33 (0.55)	<0.001
SFA (g)	24 (12)	23 (12)	24 (13)	22 (10)	28 (12)	26 (11)	0.276	25 (11)	24 (12)	0.902
MUFA (g)	24 (11)	25 (13)	24 (11)	23 (10)	26 (11)	24 (14)	0.323	22 (9)	24 (11)	0.160
PUFA (g)	12 (6)	13 (7)	11 (6)	12 (5)	13 (8)	13 (8)	0.100	11 (4)	13 (7)	0.023
Calcium (mg)	968(345)	976 (263)	985 (339)	1010 (318)	1025 (417)	967 (452)	0.284	887 (254)	982 (359)	0.241
Zinc (mg)	9.44 (3.21)	9.33 (4.31)	9.68 (3.58)	9.23 (2.83)	9.82 (2.83)	10.34 (3.73)	0.790	9.57 (3.44)	9.43 (3.26)	0.859
Iodine (µg)	177 (79)	169 (78)	168 (71)	189 (80)	186 (89)	166 (99)	0.421	150 (70)	180 (78)	0.015
Iron (mg)	12.1 (3.9)	12.9 (5.7)	11.7 (4.3)	11.9 (3.6)	12.5 (4.1)	11.6 (4.7)	0.877	12.5 (3.9)	12.1 (4.0)	0.900
Selenium (µg)	65 (31)	63 (35)	65 (28)	64 (29)	71 (41)	61 (39)	0.999	62 (32)	66 (32)	0.195
Potassium (mg)	3955 (1217)	3882 (1573)	3897 (1131)	3932 (1036)	4318 (1567)	4064 (1254)	0.302	3843 (1129)	3977 (1243)	0.281
Phosphorus (mg)	1506 (486)	1502 (508)	1499 (483)	1496 (495)	1656 (512)	1544 (366)	0.501	1447 (368)	1508 (488)	0.151
Niacin (mg)	24 (9)	22 (12)	23 (8)	24 (7)	25 (11)	25 (10)	0.567	25 (8)	24 (9)	0.523
Vit B12 (µg)	8.19 (5.24)	7.46 (5.84)	8.23 (5.08)	8.38 (4.69)	8.28 (6.26)	8.21 (5.22)	0.924	7.00 (5.35)	8.31 (5.27)	0.449
Vit C (mg)	140 (74)	150 (97)	142 (64)	132 (77)	143 (88)	130 (63)	0.703	134 (88)	142 (72)	0.520
Vit E (mg)	12.2 (5.1)	12.1 (4.6)	11.2 (5.3)	12.2 (4.7)	12.9 (5.7)	12.1 (6.7)	0.059	10.6 (4.1)	12.2 (5.3)	0.047
Vit B2 (mg)	2.12 (0.75)	2.09 (0.80)	1.99 (0.64)	2.15 (0.74)	2.31 (0.86)	2.34 (0.58)	0.063	2.0 (0.62)	2.16 (0.77)	0.163
Vit B6 (mg)	2.37 (0.89)	2.44 (1.11)	2.37 (0.92)	2.32 (0.80)	2.46 (0.97)	2.36 (1.02)	0.826	2.35 (0.97)	2.39 (0.88)	0.722
Vit B1 (mg)	1.52 (0.50)	1.47 (0.62)	1.50 (0.51)	1.51 (0.46)	1.64 (0.60)	1.47 (0.54)	0.480	1.44 (0.47)	1.56 (0.50)	0.308
Vit D2 (µg)	3.22 (3.15)	2.86 (3.21)	3.27 (3.08)	3.08 (3.07)	3.92 (3.20)	2.92 (2.66)	0.984	2.79 (3.15)	3.30 (3.15)	0.187
Vit A-retinol (µg)	346 (700)	300 (257)	359 (779)	333 (753)	342 (401)	636 (814)	0.338	387 (753)	344 (7688)	0.589
Vit A- retinol equivalents (µg)	1264 (899)	1257 (848)	1259 (868)	1266 (769)	1242 (1053)	1361 (936)	0.868	1296 (1078)	1259 (870)	0.403
Sodium (mg)	2525 (1000)	2462 (1088)	2549 (924)	2496 (1016)	2680 (1003)	2546 (1372)	0.380	2463 (752)	2545 (1020)	0.313
Magnesium (mg)	348 (109)	372 (134)	343 (95)	344 (96)	379 (127)	335 (95)	0.520	332 (96)	353 (110)	0.066
Chloride (mg)	3889 (1463)	3764 (1491)	3876 (1341)	3822 (1606)	4076 (1429)	3939 (1856)	0.349	3666 (972)	3935 (1493)	0.218
Manganese (mg)	3.83 (1.64)	4.08 (2.08)	3.83 (1.67)	3.84 (1.32)	4.13 (1.74)	3.23 (1.05)	0.190	3.30 (1.0)	3.97 (1.59)	0.009
Copper (mg)	1.22 (0.56)	1.30 (0.60)	1.18 (0.56)	1.21 (0.48)	1.19 (0.57)	1.20 (0.78)	0.804	1.12 (0.57)	1.22 (0.54)	0.305
Folate (µg)	330 (113)	359 (150)	323 (111)	328 (115)	352 (123)	333 (109)	0.681	331 (119)	331 (116)	0.480
Nitrogen (g)	13.8 (5.0)	13.1 (6.5)	13.9 (4.9)	13.9 (4.7)	14.4 (5.0)	13.8 (5.4)	0.868	13.8 (5.1)	13.8 (5.0)	0.776
Carotene (µg)	4449 (2807)	4785 (3029)	4724 (2838)	4034 (3007)	4951 (3455)	4692 (1788)	0.329	4886 (2896)	4410 (2819)	0.753

Abbreviations: %TEI, proportion in total energy intake; MUFA, Monounsaturated fatty acids; PUFA, Polyunsaturated fatty acids; SFA, Saturated fatty acids; Vit, Vitamins.

**Table 3 nutrients-13-03076-t003:** Dietary food items intake pattern of participants according to age group and muscle mass status.

Food Items	All Participants *n* = 281	60–64 Years *n* = 39	65–69 Years *n* = 98	70–74 Years *n* = 94	75–79 Years *n* = 34	80–91 Years *n* = 16	*p*-Value (Age Groups)	Low Relative Skeletal Muscle Mass *n* = 37	High Relative Skeletal Muscle Mass *n* = 244	*p*-Value (Relative Skeletal Muscle Mass Groups)
Cereals and its products (g)	212 (125)	249 (122)	218 (134)	205 (119)	217 (104)	172 (119)	0.206	186 (113)	216 (133)	0.140
Milk and milk products (g)	416 (232)	388 (301)	363 (176)	432 (231)	465 (288)	436 (263)	0.087	347 (149)	426 (264)	0.035
Eggs and egg dishes (g)	22 (18)1	22 (15)	22 (15)	22 (19)	22 (25)	22 (20)	0.545	22 (25)	22 (15)	0.342
Fats and oils (g)	13 (13)	12 (12)	13 (14)	13 (9)	16 (17)	15 (12)	0.345	13 (15)	13 (12)	0.864
Fish and fish products (g)	57 (60)	57 (57)	55 (57)	68 (61)	54 (72)	46 (55)	0.887	35 (46)	61 (58)	0.006
Fruit (g)	276 (199)	269 (266)	271 (168)	278 (213)	318 (209)	208 (238)	0.410	224 (199)	279 (192)	0.133
Meat and meat products (g)	87 (77)	76 (111)	85 (89)	90 (65)	80 (45)	116 (67)	0.142	110 (87)	83 (74)	0.007
Nuts and seeds (g)	4 (21)	13 (22)	4 (24)	4 (13)	4 (30)	2 (19)	0.283	2.1 (12.9)	4.2 (21.6)	0.005
Potatoes (g)	71 (60)	63 (48)	71 (48)	71 (48)	79 (80)	73 (85)	0.235	74.5 (49.2)	71.4 (62.8)	0.932
Soups and sauces (g)	55 (70)	51 (34)	49 (77)	56 (77)	75 (97)	53 (75)	0.304	62 (90)	55 (69)	0.785
Vegetables (g)	339 (217)	333 (226)	353 (219)	318 (202)	338 (245)	341 (185)	0.451	370 (318)	335 (196)	0.793
Sugars; preservatives and snacks (g)	24 (26)	23 (29)	22 (27)	24 (27)	24 (24)	28 (46)	0.765	25 (33)	23 (24)	0.972

**Table 4 nutrients-13-03076-t004:** Multivariable linear regression analyses showing the association of established dietary protein intake threshold with body composition and muscle phenotypes.

Outcome Variables	Unstandardized Coefficients (B) Protein Threshold	95% C.I	*p*-Value	Partial Correlations
BMI (kg/m^2^)	−3.877	(−4.933; −2.821)	<0.001	−0.399
Fat mass (kg)	−7.836	(−9.751; −5.922)	<0.001	−0.437
Biceps brachii (cm)	−0.155	(−0.235; −0.074)	<0.001	−0.223
HGS/Body mass (kg/kg)	0.063	(0.039;0.087)	<0.001	0.299

Abbreviations: BMI, Body mass index; HGS, Handgrip strength. All the analyses were adjusted for age, energy intake and physical activity level.

## Data Availability

The data used in the present study are available from reasonable request from corresponding author.

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
