# Peer review of "Dietary Protein Requirement Threshold and Micronutrients Profile in Healthy Older Women Based on Relative Skeletal Muscle Mass"

_nutrients, 2021, doi:10.3390/nu13093076_

Round 1
Reviewer 1 Report
Thank you for submitting this article which presents an area of high interest for me.
I have some comments and questions for you.
- The use of the term “elderly” starting in the introduction if pejorative and vague. If your recruited participants start at age 60, your participants are not “elderly” at all. Consider your references numbered 70 to 72—see “older adults”, or “post-menopausal women” if applicable; “older persons”
- Throughout the manuscript, terms such as “muscle quality” and “relative skeletal muscle mass” seem to be used interchangeably. These terms are not the same; how do you define muscle quality? What are your references? It is my understanding that muscle quality has yet to be defined, at least in consensus.
- Why did you recruit (or use) only females? What is your rationale?
- Did you control for PA when determining associations/cut points for needed protein, and if so, why or why not?
- Please give more detailed methods regarding determination for cut point for protein. Seems like you ended up with very uneven groups; wouldn’t this be a limitation to interpretation of your results and conclusions? The older group is the smallest so not sure if the results from this small group can be trusted. Was supplement information (e.g. protein supplements, vitamin/minerals, etc.) collected as part of the FFQ?
- In section 4.1, you discuss “undernourished” as a risk factor for low muscle mass. Were your participants undernourished, which ones, and what was that based on (energy per kg?) The reference number 54 is obsolete; please find updated reference, same source and see how they define undernourished.
- Section 4.2, last paragraph, last sentence seems to be missing some rationale? Which chronic disease? Do all individuals aged 60 and older have chronic disease?
- PUFA is not a micronutrient
- How many from each age group make up the small n=37 low relative muscle mass group? Were all the aged 80-91 in this n=37?
Author Response
Reviewer 1 comments:
Thank you for submitting this article which presents an area of high interest for me.
I have some comments and questions for you.
- The use of the term “elderly” starting in the introduction if pejorative and vague. If your recruited participants start at age 60, your participants are not “elderly” at all. Consider your references numbered 70 to 72—see “older adults”, or “post-menopausal women” if applicable; “older persons”.
Author’s response: We do agree with the reviewer’s comments on the use of terminology “elderly” for the population selected in the present study (i.e., 60+ years). We have therefore changed the word “elderly” to older population/older adults throughout the manuscript in the revised version as per reviewer suggestion.
- Throughout the manuscript, terms such as “muscle quality” and “relative skeletal muscle mass” seem to be used interchangeably. These terms are not the same; how do you define muscle quality? What are your references? It is my understanding that muscle quality has yet to be defined, at least in consensus.
Author’s response: As per the reviewer’s comment, we have checked the entire manuscript, and used the terminologies “relative muscle mass” and “muscle quality” consistently in the revised version. While doing so, we have made the meaning of these terminologies clearer as – muscle quality defined as Hand grip strength/Body mass, and relative muscle mass as 100 * skeletal muscle mass/Body mass) throughout. Furthermore, wherever we indicated relative muscle mass, it is abbreviated as %SMM.
We do agree with the reviewer that there is little consensus for defining muscle quality, however, we define muscle quality similarly to several previous studies that defined it as muscle strength relative to body mass or lean/skeletal muscle mass.
- Why did you recruit (or use) only females? What is your rationale?
Author’s response: The recruitment of females only in the present study is based on the findings from previous literature that men and women have different rates of muscle mass/strength reduction with ageing, and importantly the values/quantity of skeletal muscle mass and strength are in different ranges between the two sexes at any age. Therefore, we believed that the consideration of one sex only in a study gives a clearer and more accurate measure of protein requirement with proportion of muscle mass present.
- Did you control for PA when determining associations/cut points for needed protein, and if so, why or why not?
Author’s response: There was no association between relative muscle mass and PA (Spearman correlation coefficient (rho) = 0.72, p = 0.226) (not shown in manuscript) and PA did not impact the outcome threshold measure while performing ROC analysis (not shown in manuscript). As considering variables not significantly associated while establishing a threshold/cut-off is not recommended, we did not consider PA at least while establishing threshold. However, we do mention the limitation of the PASE questionnaire to estimate PA level in the limitations part of the Discussion.
- Please give more detailed methods regarding determination for cut point for protein. Seems like you ended up with very uneven groups; wouldn’t this be a limitation to interpretation of your results and conclusions? The older group is the smallest so not sure if the results from this small group can be trusted. Was supplement information (e.g. protein supplements, vitamin/minerals, etc.) collected as part of the FFQ?
Author’s response: We have a brief description about how the protein threshold was established in the Statistics section in the revised version and it reads as: “The minimally required protein threshold was defined as the value that had both higher sensitivity and specificity values……………..”
In our response to the reviewer’s other comment (no.8), we can see that lower muscle mass prevalence is not concentrated in the older age categories. Therefore, we believe that the derived minimal protein threshold for the maintenance of higher muscle mass is applicable to the whole sample, not only the oldest group.
We appreciate the reviewer comment regarding supplement intake. As we have no information of any supplement intake from older population, we have added this in the limitation in the revised version:
- In section 4.1, you discuss “undernourished” as a risk factor for low muscle mass. Were your participants undernourished, which ones, and what was that based on (energy per kg?) The reference number 54 is obsolete; please find updated reference, same source and see how they define undernourished.
We appreciate the feedback regarding the term originally used in section 4.1 and have changed “undernutrition” to “lower nutrient intake”. Since the content may be shadowed by addition of undernutrition, we have deleted the reference no 54 in this version.
- Section 4.2, last paragraph, last sentence seems to be missing some rationale? Which chronic disease? Do all individuals aged 60 and older have chronic disease? PUFA is not a micronutrient
Author’s response: In the first paragraph of section 4.2, we have the sentence “Given that during acute and chronic diseases, an even higher protein intake (1.2 g/kg/d) is recommended in older adults, we suggest that an RDA of 0.75 or 0.8 g/kg/d is insufficient to sustain muscle health in older women.” in the revised version. Here, we changed “may be required” to “recommended” as it is an evidence-based recommendation paper.
We have all the participants, who were physically independent, and their activities of daily livings were not affected by any of the acute/chronic condition.
In the revised submitted version, we have removed PUFA from the list of micronutrients and added it separately. Thank you.
- How many from each age group make up the small n=37 low relative muscle mass group? Were all the aged 80-91 in this n=37?
Author’s response: The 37 with low relative muscle mass are grouped in age categories below:
60-64 years=3
65-69 years=14
70-74 years=14
75-79 years=2
80-91 years=4
Our data shows that prevalence of lower muscle mass is 37/281 =13.2% and the prevalence of lower muscle mass in oldest category is 4/16=25%. This indicates that there is dominance of other age groups as well (not the oldest age groups only).
Reviewer 2 Report
The article by Khanal et al, presented a study on the comparison of dietary intakes of elderly women between 60-91 years old, with low and high muscle mass. They provided details of intakes of microminerals and food items in relation to age group and muscle mass capacity finally suggesting the minimal dietary intake of protein for proper maintenance of muscle mass. The objectives are clearly presented and the findings will benefit the readers of Nutrients.
Minor comments:
- The last sentence in the abstract is not a conclusion from this study so it is not necessary.
- Table 1. Please, describe the meaning of the numbers shown as superscripts in the various age group columns
- Sometimes Vitamin E/Vit E is used in the text, it is better to stick to a single format
Author Response
Minor comments:
- The last sentence in the abstract is not a conclusion from this study so it is not necessary.
Author’s comment: We have deleted the last sentence from the abstract as per the reviewer’s suggestion in the revised version. Thank you.
- Table 1. Please, describe the meaning of the numbers shown as superscripts in the various age group columns
Author’s comment: Thank you for noting this in our submitted version. We have added the meaning of superscript in the age group columns as following in the revised version:
“1,2,3,4,5 indicate difference from 60-64 years, 65-69 years, 70-74 years, 75-79 years and 80+ years, respectively at p≤0.05.”
- Sometimes Vitamin E/Vit E is used in the text, it is better to stick to a single format.
Author’s response: Thank you. We have used Vit E consistently in the revised version.